

# Implications of dominance hierarchy on hummingbird-plant interactions in a temperate forest in Northwestern Mexico

Gabriel López-Segoviano[1], Maribel Arenas-Navarro[2], Laura E. Nuñez-Rosas[3] and María del Coro Arizmendi[3]

[1] Escuela Nacional de Estudios Superiores (ENES), Unidad Morelia, Universidad Nacional Autónoma de México, Morelia, Michoacán, México
[2] Instituto de Investigaciones en Ecosistemas y Sustentabilidad (IIES), Universidad Nacional Autónoma de México, Morelia, Michoacán, Mexico
[3] Facultad de Estudios Superiores Iztacala, Universidad Nacional Autónoma de México, Tlalnepantla, Estado de México, Mexico

Corresponding author
María del Coro Arizmendi,
coro@unam.mx

## ABSTRACT

The structuring of plant-hummingbird networks can be explained by multiple factors, including species abundance (*i.e.*, the neutrality hypothesis), matching of bill and flower morphology, phenological overlap, phylogenetic constraints, and feeding behavior. The importance of complementary morphology and phenological overlap on the hummingbird-plant network has been extensively studied, while the importance of hummingbird behavior has received less attention. In this work, we evaluated the relative importance of species abundance, morphological matching, and floral energy content in predicting the frequency of hummingbird-plant interactions. Then, we determined whether the hummingbird species' dominance hierarchy is associated with modules within the network. Moreover, we evaluated whether hummingbird specialization (*d'*) is related to bill morphology (bill length and curvature) and dominance hierarchy. Finally, we determined whether generalist core hummingbird species are lees dominant in the community. We recorded plant-hummingbird interactions and behavioral dominance of hummingbird species in a temperate forest in Northwestern Mexico (El Palmito, Mexico). We measured flowers' corolla length and nectar traits and hummingbirds' weight and bill traits. We recorded 2,272 interactions among 13 hummingbird and 10 plant species. The main driver of plant-hummingbird interactions was species abundance, consistent with the neutrality interaction theory. Hummingbird specialization was related to dominance and bill length, but not to bill curvature of hummingbird species. However, generalist core hummingbird species (species that interact with many plant species) were less dominant. The frequency of interactions between hummingbirds and plants was determined by the abundance of hummingbirds and their flowers, and the dominance of hummingbird species determined the separation of the different modules and specialization. Our study suggests that abundance and feeding behavior may play an important role in North America's hummingbird-plant networks.

## INTRODUCTION

Ecological network theory allows the comparison of interactions among highly diverse ecological communities and provides methods to quantify and compare interaction patterns across communities (*Bascompte & Jordano, 2006*, *2007*). Network structure can result from the simultaneous influence of species abundance and the constraints imposed by complementarity in species phenotypes, phenologies, spatial distributions, phylogenetic relationships, and sampling artifacts (*Vázquez, Chacoff & Cagnolo, 2009*). As a result, two main hypotheses have been postulated as the main factors modulating the occurrence of plant-pollinator interactions (*Sazatornil et al., 2016*). The neutrality hypothesis refers to the presumption that species abundances drive interaction frequencies, with more abundant species having more interaction partners and higher interaction frequencies than rarer species (*Vázquez et al., 2007*, *2009*; *Krishna et al., 2008*; *Vázquez, Chacoff & Cagnolo, 2009*; *Sazatornil et al., 2016*; *Simmons et al., 2021*). In contrast, the forbidden links hypothesis postulates that network patterns are constrained by morphological trait-matching, phenologies, spatial distributions, and phylogenetic relationships of plants and pollinators (*Jordano, Bascompte & Olesen, 2003*; *Vázquez, Chacoff & Cagnolo, 2009*; *Vizentin-Bugoni, Maruyama & Sazima, 2014*; *Sazatornil et al., 2016*; *Simmons et al., 2021*). Thus, two species cannot interact if their phenologies do not overlap or do not match morphologically (*Vázquez, Chacoff & Cagnolo, 2009*; *Vizentin-Bugoni, Maruyama & Sazima, 2014*). For example, phenological overlap is an important driver of hummingbird-plant assemblages in some places in Mexico (*Martín González et al., 2018*; *Chávez-González et al., 2020*). In seasonal regions, for instance, phenological overlap has a stronger influence on interactions and may be driven by hummingbirds' migratory behavior (*Sonne et al., 2020*). Thus, migrating hummingbirds are unlikely to involve one-to-one interactions with plants, because no temperate zone hummingbird species can afford to rely entirely on a single plant species for nectar (*Abrahamczyk & Renner, 2015*); which in turn leads to high seasonal turnover in hummingbird species composition.

Trait matching between hummingbird's bill traits and flowers' corolla traits is a primary driver for partitioning resources in hummingbird-plant interactions (*Wolf & Stiles, 1989*; *Maglianesi et al., 2014*; *Sonne et al., 2019*). This morphological matching reflects coevolutionary processes that are understood as increasing species' coexistence and decreasing competition for floral resources and pollinators (*Wolf & Stiles, 1989*; *Cotton, 1998a*). In this context, specialization (species that interacts with a relatively small number of the available partners; *Vázquez & Aizen, 2005*) and modularity (interactions organized in sets of interacting species or modules; *Olesen et al., 2007*) can be related to trait matching in hummingbird-plant networks (*Sonne et al., 2020*; *Dalsgaard et al., 2021*). However, matching between hummingbirds' bills and the flowers' corollas declines along latitudinal and altitudinal gradients, where strong seasonality in terms of temperature makes phenological overlap the main driver of hummingbird-plant interaction frequencies (*Sonne et al., 2020*). Accordingly, hummingbird communities of North America have less diverse bill morphology than communities in tropical environments (*Stiles, 1981*; *Kodric-Brown et al., 1984*; *Brown & Bowers, 1985*), leading to less specialized and less modular networks (*Dalsgaard et al., 2011*; *Sonne et al., 2020*).

In the Mexican hummingbird-plant assemblage, migratory behavior is apparently as important as hummingbird and flower morphological traits for structuring interactions at the community level (*Martín González et al., 2018*; *Chávez-González et al., 2020*). In addition to morphological restrictions, inter and intraspecific competition could be important factors for structuring hummingbird-plant networks (*Feinsinger, 1976*; *Wolf, Stiles & Hainsworth, 1976*; *Maglianesi, Böhning-Gaese & Schleuning, 2014*). For example, some studies have focused on the importance of dominance hierarchy in network structure and its relationship with core-periphery species (*Dworschak & Blüthgen, 2010*; *Dáttilo, Díaz-Castelazo & Rico-Gray, 2014*).

Dominance hierarchy among species can affect the visitation frequency of a hummingbird to a specific plant species (*Wolf, Stiles & Hainsworth, 1976*; *López-Segoviano, Bribiesca & Arizmendi, 2018*). The strength of species interactions plays an essential role in determining the specialization, modularity, and core-periphery status (*Blüthgen, Menzel & Blüthgen, 2006*; *Dormann & Strauss, 2014*; *Martín González et al., 2020*). Plants visited by hummingbirds and the intensity of their visits determine various metrics of the interaction networks. According to *Dormann & Strauss (2014)*, the strength of links as quantitative information is more sensitive and specific to detect modules. Likewise, the strength of the interactions is an essential piece in the calculation of specialization (*Blüthgen, Menzel & Blüthgen, 2006*) and core-periphery in the networks (*Martín González et al., 2020*). Hence, dominance hierarchy could influence the structure of hummingbird-plant networks and determine hummingbird specialization (*Ramírez-Burbano et al., 2022*). The existence of a dominance hierarchy may lead to differential utilization of plants by hummingbirds depending on the hummingbird's place in the hierarchy and the caloric value of the plant's nectar, with the most dominant hummingbird species feeding mainly on plant species that offer more calories per flower (*López-Segoviano, Bribiesca & Arizmendi, 2018*). Therefore, dominant hummingbird species can act as specialized and the less dominant species as generalists in the hummingbird-plant network.

In a temperate forest located in northwestern Mexico—El Palmito, Sinaloa—14 hummingbird species coexist during fall and winter, when there is a high abundance of flower resources and several migratory hummingbird species arrives (*López Segoviano, 2018*; *López-Segoviano et al., 2018*, *2021*). At El Palmito, the hummingbird community is structured by aggressive dominance relationships associated with the quality of floral resources (*López-Segoviano, Bribiesca & Arizmendi, 2018*). Therefore, in this work, we assessed the dominance hierarchy of hummingbird species and its influence on species' roles (specialization), as well as its association with module separation within the hummingbird-plant visitation network. To do this, we first evaluated the relative importance of species abundance, morphological match, and nectar energy of visited plants in predicting the frequency of interactions. Then, we determined whether the hummingbird species' dominance hierarchy is associated with modules within the network. Moreover, we evaluated whether hummingbird specialization (*d'*) is related to bill morphology (bill length and curvature) and dominance hierarchy. Finally, we determine whether generalist core hummingbird species are lees dominant in the community.

## METHODS

### Study area

The study site is located in the Sierra Madre Occidental near the community of El Palmito, Concordia (23°35′20N, 105°52′0W) in Sinaloa, Mexico, between 1,800 and 2,200 m asl. The climate is temperate sub-humid, with an average annual precipitation of 1,247 mm (*SMN, 2016*). Vegetation is constituted mainly of pine-oak, oak-pine, and pine forest, cloud forest, and riparian and secondary vegetation (*Ávila-González et al., 2019*). Fieldwork was conducted from 10 November 2010 to 24 February 2011 in a 300 ha area presenting a mixture of these vegetation types. The largest number of hummingbird species and floral resources are present in El Palmito during the fall/winter season (*López Segoviano, 2018*).

### Hummingbird interactions

The quantitative hummingbird–plant interaction network was constructed based on observations of legitimate visits (when hummingbirds contacted the flowers' reproductive structures). We considered the number of visits to be a measure of the strength of the interactions. A visit was defined as when a hummingbird probed at least one flower on an individual plant species (*Maruyama et al., 2019*). We quantified the visitation rate and interactions among hummingbirds by conducting observations at a distance of ~8 m from the plant species (following *Cotton, 1998b*). We observed 338 h of hummingbirds visits to ten plant species (*Salvia elegans*, *S. gesneriiflora*, *S. iodantha*, *S. mexicana*, *Cuphea watsoniana*, *C. pinetorum*, *Loeselia mexicana*, *Cestrum thyrsoideum*, *Castilleja arvensis* and *Agave inaequidens* subsp. *barrancensis*). The number of observed hours for each plant species was proportional to its abundance in the study area (described below). Behavioral observations were performed daily by two observers at 1-h intervals between 07:00 and 15:00 h, recording the hummingbird species, visitation to flowers, and the outcomes of aggressive interactions between hummingbirds. We considered aggressive interactions when hummingbird chase and attack other hummingbirds (*Kodric-Brown et al., 1984*; *Cotton, 1998b*; *Camfield, 2006*).

### Hummingbird and flowers abundance

To evaluate the abundance of hummingbirds and the flowers they visited, we counted hummingbirds in fixed-radius plots (25 m radius) and flowers along transects measuring 40 m in length and 5 m in width within each fixed-radius plot (*López-Segoviano et al., 2018*). We established 20 plots separated by at least 188 m, which were fixed and distributed to represent the heterogeneity of the study site (*López-Segoviano et al., 2018*). During 10 min, two observers identified and counted all hummingbirds visually or acoustically detected in a plot (*Tinoco, Santillán & Graham, 2018*); when the bird survey was finished, we counted all flowers in a transect. We excluded hummingbirds flying over the fixed-radius plot to avoid an overestimation of hummingbird abundance. All plots and transects were sampled every 10 days from 12 November to 20 February in 2010–2011, resulting in 11 samples.

## Hummingbird and flowers traits

We measured bill morphological traits (bill length and curvature) in live-caught hummingbirds and museum specimens housed at the Museum of Zoology 'Alfonso L. Herrera' (MZFC, UNAM) and Colección Nacional de Aves (Instituto de Biología, UNAM, Mexico City, Mexico). Three standard mist nets (12 × 3 m) were placed near flowering sites to capture hummingbirds during fieldwork. Mist nets were operated for an 8 h period beginning at sunrise during 4 days in each sampling period. Ten sampling periods were conducted from 12 November 2010 to 26 February 2011. Following *Maglianesi et al. (2014)*, we measured bill length and curvature, since these morphological traits have been found to affect plant–hummingbird interactions. Bill length (mm) of each captured individual was measured using a calliper, and bill curvatures (°) was calculated with the angle tool using ImageJ software (http://rsbweb.nih.gov/ij/) from lateral photographs from each individual.

We measured flower morphology (corolla's length, diameter, and curvature) and nectar characteristics (concentration, volume, and calories) of plant species visited by hummingbirds (*Maglianesi et al., 2014*). To calculate corolla curvatures (°), we used the same method of bill curvature. We quantified the mean sugar production per flower for each species by measuring the nectar produced and sugar concentration from recently opened flowers that were bagged for 24 h (*Tinoco, Santillán & Graham, 2018*). Nectar was extracted using microcapillary tubes, and nectar concentration was measured using a portable refractometer (0–32 Brix). Calories produced per flower was calculated by multiplying nectar volume (μL) by sugar concentration (mol) by 1.34, as proposed by *Stiles (1975)*. We calculated mean values for each of the ten plant species using 11–134 individuals (depending on their abundance).

## Dominance hierarchy

We identified the winner of an aggressive interaction as the hummingbird that returned to feed or perch after it had successfully defended and/or chased off another hummingbird from a floral patch (*Justino, Maruyama & Oliveira, 2012*). We considered the hummingbird lost when it was aggressively displaced from the floral patch by another hummingbird. We calculated the hummingbird species' dominance hierarchy using David's score ($Ds = w + w^2 - l - l^2$). This score reflects the proportion of wins by species $i$ in its interactions with species $j$, as $w + w^2 - l - l^2$, where $w$ is the number of $i$'s wins, $l$ is $i$'s losses, $w^{(2)}$ is the wins of species defeated by $i$, and $l^{(2)}$ is the losses of species to whom $i$ lost (*David, 1987*; *de Vries, 1998*). This index for ranking dominance was designed for an incomplete data matrix, with paired comparisons in which not all species compete against each other (*David, 1987*). The resulting David's scores indicate the dominance rank of each hummingbird species within the interspecific interaction matrix (*López-Segoviano, Bribiesca & Arizmendi, 2018*), in this case, 13 hummingbird species.

We also calculated Dominance certainty (DC) is a method that utilizes both direct and indirect information *via* social network analysis to gauge the overall "fit" of an individual's position in the dominance network (*Fushing et al., 2011*; *Linden et al., 2019*; *McCowan et al., 2022*). DC was calculated from dominance interactions using the percolation and

conductance method implemented *via* the Perc package (*Fujii et al., 2015*; *Vandeleest et al., 2016*; *McCowan et al., 2022*) in R software 4.2.3 (*R Core Team, 2023*). Perc package computes a network-based ranking method that combines information from direct dominance interactions with information from multiple indirect dominance pathways (*via* common third parties) to quantify dyadic dominance relationships and yield ordinal ranks from such relationships (*Fushing et al., 2011*; *McCowan et al., 2022*). Therefore, species with high DC indicate that they are more certain of their rank position than individuals with low DC. We then standardized ordinal ranks to account for group size and created a rank index ranging between zero and one, indicating the top- and bottom-ranking hummingbird, respectively. Values typically range from 0.5 (uncertain position; an individual has an equal probability of ranking higher or lower than another individual) to 1 (certain position; an individual has a 100% probability of ranking higher than another individual; *Fushing et al., 2011*; *McCowan et al., 2022*). Because we recorded few interactions of the Ruby-throated hummingbird (*Archilochus colubris*) and Broad-tailed hummingbird (*Selasphorus platycercus*) species and a DC of 0.5 was obtained for both species, we decided to regenerate the matrix omitting these species to have a more robust interaction matrix.

## Network analysis

### Specialization

To calculate the degree of specialization (*d'*) of hummingbirds and plant species within the interaction network, we used the function "*specieslevel*" in the *bipartite* package (*Blüthgen, Menzel & Blüthgen, 2006*; *Dormann, Gruber & Fründ, 2008*). The *d'* value is derived from the Kullback-Leibler distance; it measures the deviation of the actual interaction frequencies from a null model that assumes that all partners interact in proportion to their availability (*Blüthgen, Menzel & Blüthgen, 2006*). The *d'* value ranges from 0 for the most generalized to 1 for the most specialized species (*Blüthgen, Menzel & Blüthgen, 2006*). The *d'* values were weighted by total interaction frequencies of hummingbirds (square root transformed) following *Maglianesi et al. (2014)*.

### Core-periphery

To determine the hummingbird species' core and periphery roles within the interaction network, we used the function "*CPness*" in the *econetwork* package (*Miele et al., 2020*; *Martín González et al., 2020*; *Miele, Ramos-Jiliberto & Vázquez, 2020*). The *CPness* function displays a matrix of core-periphery structure, and there is a species ordering such that interactions are distributed in an L-shape composed of four blocks of varying connectance (see *Martín González et al., 2020*; *Miele et al., 2020*). The core-periphery species was determined by $CPness = (E11 + E12 + E21)/E$, where Eij is the number of interactions (edges) or the sum of weights for each block (Eij for block ij) or for the entire network (E) (*Miele et al., 2020*; *Martín González et al., 2020*).

### Modularity

To calculate the modularity (*Q*) within the network we used the Beckett algorithm (DIRTLPAwb+) for weighted (quantitative) bipartite networks
_________________________________________________________________

(*Dormann & Strauss, 2014*), for maximized weighted modularity in bipartite networks (*Beckett, 2015*). The modularity (*Q*) measures aggregated sets of interacting species within the network, ranging from 0 to 1 (*Dormann & Strauss, 2014*). Modularity values are highest when each module is isolated from the rest of the network (*Beckett, 2015*). We used the number of Markov chain Monte Carlo (MCMC) moves to yield no improvement before the algorithm stops set to $10^7$ steps (*Dormann & Strauss, 2014*). Higher *Q* values indicate that the data support the division of a network into modules. Following *Maruyama et al. (2014)*, we performed 50 independent runs and retained the module conformation with the highest *Q* value. Because the algorithm is stochastic, the module arrangement can vary between each run, so we evaluated whether the *Q* value of each network was different than expected by chance, performing a null model ("Patefield algorithm") for *Q* values with 100 randomly generated network replicates using the observed species richness and interaction heterogeneity. We used the Q values in the randomizations to calculate the z-score, which is the number of standard deviations a datum is above the mean of the 100 randomized networks (*Maruyama et al., 2014*).

### Constructing the model to predict the interactions frequency

To evaluate which factors contributed to the structure of the observed flower-hummingbird interactions, we followed *Benadi et al. (2021)*, who used a generalized additive model (GAM) in the mgcv package (*Wood & Wood, 2015*). This method is flexible to include any function translating trait matches into interaction probabilities (*Benadi et al., 2021*). We fitted a negative-binomial GAM with gamma to 1.4 to prevent over-fitting, using 1D splines for the observed trait values per species of each group (hummingbirds and flowering plants) and their matching (*Benadi et al., 2021*). The option gamma = 1.4 helps us to avoid overfitting by the smoothers and puts a heavier penalty on each degrees of freedom in the generalised cross-validation score (*Zuur et al., 2009*; *Wood & Wood, 2015*). We used the number of interactions of each pair of species as response variable and the proportion of abundances (hummingbirds and flower of plants), mean of calories per flower species, calories per plant species (calories per flower × total flowers), hummingbird weight, David score (Ds) of hummingbirds, morphological match, bill, and corolla length as predictors. The abundance of flowers and hummingbirds was obtained from the records at the fixed-radius plots and flower count transects. We built morphological match based on hummingbird bill length and curvature and flower corolla length and curvature: first standardizing all trait variables to zero mean and unit variance, second, the morphological match was calculated as the Euclidean distance in traits between each hummingbird–plant pair (*Weinstein & Catherine, 2017*; *Sonne et al., 2019*).

### Statistical analysis

To examine if the hummingbird and plant species were grouped into modules and which are the most important traits, we performed three principal component analyses (PCA). The first PCA was done with the hummingbird traits (David's score and weight, bill length, and curvature), the second PCA with the Perc ranking and the hummingbird traits the third PCA with the floral traits (nectar volume, concentration and calories, and corolla

**Table 1 Results from GAM analysis to assess relationships between the frequency of interactions in the network and traits of hummingbird and plant species and morphological match ($Z = -3.537$, $R^2$ adj = 0.259, *Deviance explained* = 84.2%).**

|  | edf | $X^2$ | P |
|---|---|---|---|
| Morphological match | **1.827** | **6.119** | **0.042** |
| Flowers abundance | **1** | **3.949** | **0.046** |
| Hummingbird abundance | **1.838** | **58.103** | **<0.001** |
| Calories per flower | **1.829** | **9.547** | **0.015** |
| Calories per species | 1.862 | 4.261 | 0.087 |
| Corolla length | 1 | 1.874 | 0.171 |
| Bill length | 1.183 | 0.259 | 0.658 |
| Hummingbird weight | 1 | 0.808 | 0.368 |
| David's score (Ds) | 1 | 0.044 | 0.833 |

**Note:**
Significant results are given in bold ($P < 0.05$).

length, width, and curvature) by species. To evaluate whether hummingbird species specialization (*d'*) was related to the dominance hierarchy, we performed a linear regression between the hummingbirds' dominance hierarchy (Ds and Perc ranking) and their degree of specialization (*d'*). Also, we evaluated the relationship between bill length and curvature with the degree of specialization (*d'*). We performed all analyses in R software version 4.2.3 (*R Core Team, 2023*).

# RESULTS

We recorded 2,272 interactions among 13 hummingbird species and 10 plant species (Table S1). The GAM model indicated that hummingbird abundance is the best factor to predict the frequency of interactions; also, flower abundance, and calories per flower, were associated (Table 1). Also, we found a relationship between the frequency of interactions and morphological match; hummingbirds interact more with plants with which they do not have a morphological match (Table 1).

The agonistic interaction network (477 interactions between 13 species) has 36.7 interactions per individual, 6.1 interactions per dyad, and 0.551 proportion of unknown relationships. Both rankings (Ds and Perc ranking) showed that Blue-throated Mountain-gem (*Lampornis clemenciae*) Rivoli's Hummingbird (*Eugenes fulgens*), and Mexican Violetear (*Colibri thalassinus*) were the more dominant, and, Bumblebee Hummingbird (*Selasphorus heloisa*), Calliope Hummingbird (*S. calliope*), and Costa's Hummingbird (*Calypte costae*) were the less dominant in the community (Table 2). Dominance certainty analysis shows the heatmap of dominance probabilities between hummingbird species (Fig. S1).

We found a low modularity value of the hummingbird-plant network (Q = 0.201) that was different from the null model when compared to the randomized differences of Q values (Z test = 37.40, P-values < 0.05). We found three modules: the first module contained the most dominant hummingbird species (Rivoli's Hummingbird, and Blue-throated Mountain-gem), which interacted mainly with *Agave inaequidens* flowers

**Table 2 David's score, Perc ranking, and dominance certainty (DC) for the hummingbird species.**

| Species | David's score *Ds* | Perc Ranking | Dominance certainty (DC) Mean | SD |
|---|---|---|---|---|
| Berylline Hummingbird | 11.680 | 4 | 0.917 | 0.139 |
| Blue-throated Mountain-gem | 20.905 | 1 | 0.828 | 0.175 |
| Broad-billed Hummingbird | −0.333 | 6 | 0.861 | 0.160 |
| Broad-tailed Hummingbird | −7.596 | | | |
| Bumblebee Hummingbird | −18.680 | 11 | 0.975 | 0.027 |
| Calliope Hummingbird | −13.180 | 10 | 0.911 | 0.152 |
| Costa's Hummingbird | −17.566 | 9 | 0.872 | 0.174 |
| Mexican Violetear | 17.712 | 3 | 0.843 | 0.178 |
| Rivoli's Hummingbird | 24.992 | 2 | 0.968 | 0.063 |
| Ruby-throated Hummingbird | −6.552 | | | |
| Rufous Hummingbird | −12.541 | 8 | 0.907 | 0.116 |
| Violet-crowned Hummingbird | 4.151 | 5 | 0.911 | 0.128 |
| White-eared Hummingbird | −2.990 | 7 | 0.921 | 0.126 |

**Note:**
We excluded Ruby-throated hummingbirds and broad-tailed hummingbirds from Perc ranking analysis for their low dominance certainty.

(Fig. 1 and Table 1). The second module contained three dominant hummingbirds (Berylline Hummingbird *Saucerottia beryllina*, Violet-crowned Hummingbird *Ramosomyia violiceps*, and Mexican Violetear), and three less dominant hummingbird species (Ruby-throated Hummingbird *Archilochus colubris*, Costa's Hummingbird, and Broad-billed Hummingbird *Cynanthus latirostris*) interacting mostly with *Cestrum thyrsoideum* flowers (Fig. 1 and Table 1). The third module contained five less dominant hummingbird species (White-eared Hummingbird *Basilinna leucotis*, Bumblebee Hummingbird, Calliope Hummingbird, Rufous Hummingbird, and Broad-tailed Hummingbird *S. platycercus*) that interacted mainly with flowers of *Salvia* species (Fig. 1 and Table 1).

The PCA with hummingbird traits and the Ds showed that the first two components of the PCA explained 95.9% of the total variance. The first axis (PC1) explained 73.8% of the variance and was related to the Ds (with the highest contribution; Table S2). The second axis (PC2) explained 22.2% of the variance and was related to bill curvature (with the highest contribution) and bill length (Fig. 2A). The second PCA with hummingbird traits and the Perc analysis showed that the first two components of the PCA explained 96.1% of the total variance. The first axis (PC1) explained 73.5% of the variance and was related to the Perc ranking (with the highest contribution; Table S2). The second axis (PC2) explained 22.5% of the variance and was related to bill curvature (Fig. 2B). The more dominant hummingbird species in PCA1 and PCA2 that belonged to module one were located in the positive quadrant of PC1 and negative of PC2 in both PCA's. The third PCA with the floral traits (Fig. 2C) showed that the first two components of the PCA explained 71.97% of the total variance (Table S3). The first axis explained 52.13% of the variance and

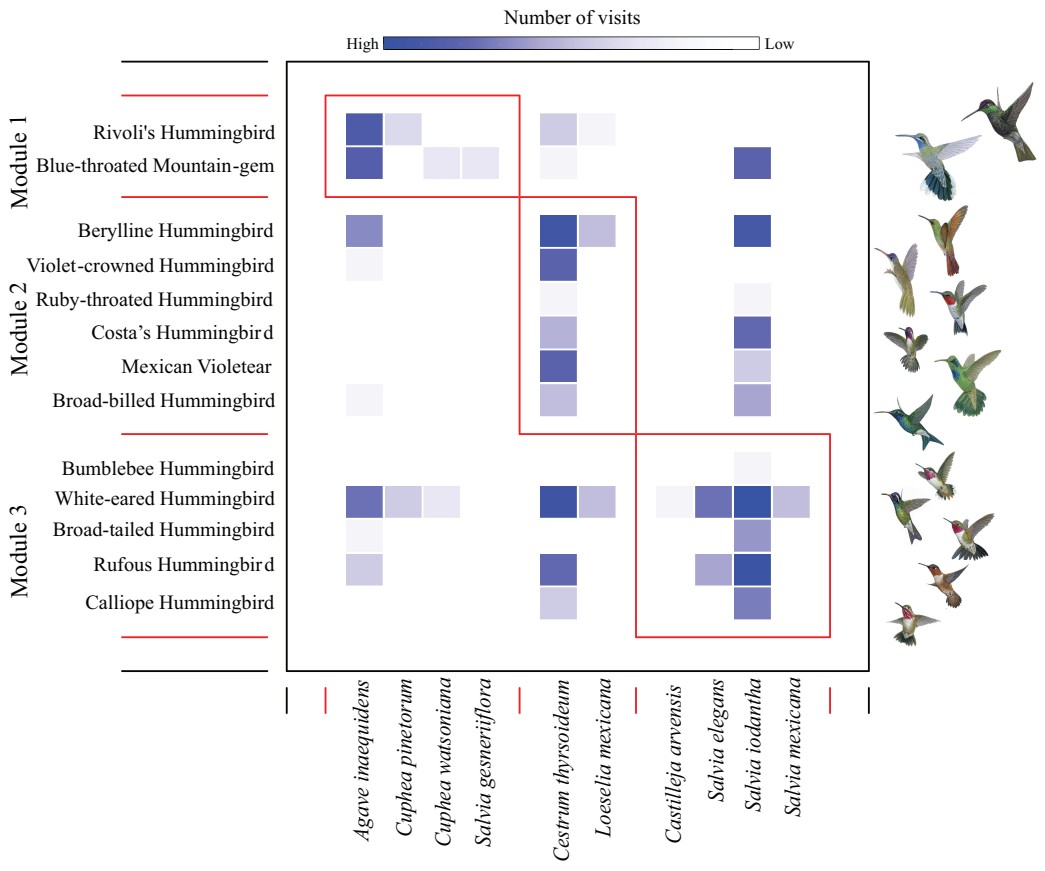

**Figure 1 Modules in the plant–hummingbird network from El Palmito during fall/winter.** The intensity of blue shading represents the interaction frequency. Hummingbird illustration credit: Marco Antonio Pineda Maldonado/Banco de Imágenes CONABIO.

was related to nectar volume (with the highest contribution) and sugar concentration (brix). The second axis explained 19.84% of the variance and was related to flower length (with the highest contribution) and width. The species that make up module three were located in the positive quadrant of PCA1 and negative of PC2; while the species that belonged to module one and two were located dispersed in the rest of the quadrants (Fig. 2C).

We found that more specialized hummingbird species were Rivoli's Hummingbird, Violet-crowned Hummingbird, and Blue-throated Mountain-gem (Table S4). The species-level network specialization ($d'$) was related to dominance (Ds: $R^2 = 0.78$, $F_{1,12} = 38.280$, $P < 0.001$ and Perc ranking: $R^2 = 0.77$, $F_{1,10} = 30.86$, $P < 0.001$; Figs. 3A and 3B), and bill length of hummingbird species ($R^2 = 0.81$, $F_{1,12} = 47.60$, $P < 0.001$; Fig. 3C), while bill curvature was not significantly associated ($R^2 = 0.12$, $F_{1,12} = 1.51$, $P = 0.244$; Fig. 3D).

White-eared and Rufous hummingbirds were identified as the generalist core species of the network (Fig. S2, and Table S4). These hummingbird species were the less dominant hummingbird species at this area (Table S4).

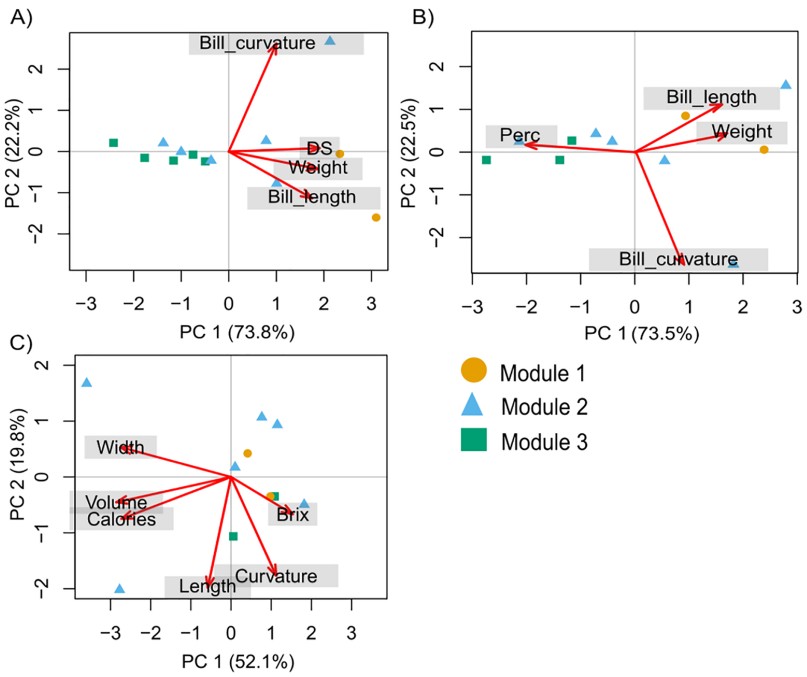

**Figure 2 Principal components analysis (PCA) for the hummingbird rankings and floral traits.**
(A) David's score (Ds) ranking, (B) Perc ranking, and (C) floral traits. The colors indicate the modules in the plant–hummingbird network.

## DISCUSSION

We found that one of the main drivers of plant-hummingbird interaction frequencies was each species' abundance, as proposed by the neutrality interaction theory (*Vázquez et al., 2009*). Highlighting, hummingbird abundance as the most important factor to predict plant-hummingbird interactions over morphological traits and the match bill-corolla. *Sonne et al. (2020)* proposed that morphological matching is a less important driver of plant-hummingbird interaction frequencies far from the equator. Biogeographical and evolutionary history has an important role in patterns of trait-matching in hummingbird–plant associations (*Dalsgaard et al., 2021*). North American hummingbird clades (Bees, Mountain Gems, and Emeralds) are less morphologically specialized than those of other clades in south America, such as the Hermits, Mangoes, and Patagona, which have longer and more curved bills (*Rodríguez-Flores et al., 2019*). Meanwhile, other studies have found that phenological overlap is the primary driver of the plant-hummingbird assemblage in Mexico (*Martín González et al., 2018*; *Chávez-González et al., 2020*). In this study, we do not include the phenological overlap to predict the frequency of hummingbird-plant interactions, because we covered the four months when the blooming of principal flower hummingbird resources influences the arrival of several migratory hummingbird species (*López-Segoviano et al., 2021*). Thus, most flowering plants and hummingbirds coincide, and future studies should focus on long-term monitoring and include plant phenology.

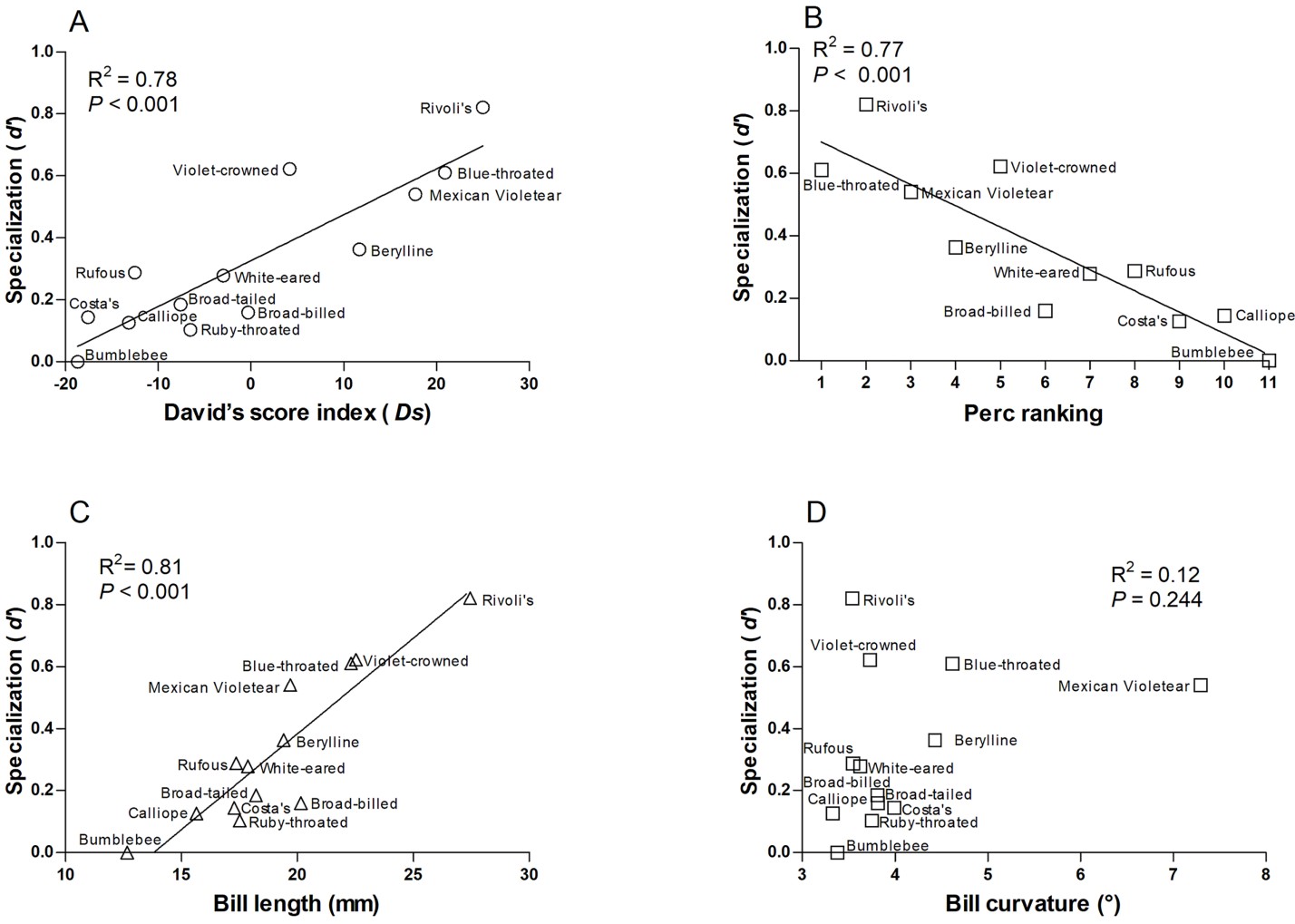

**Figure 3** Relationships of the species-level network specialization (*d'*) to dominance hierarchy (A) David's score, and (B) Perc ranking, bill length (C), and bill curvature (D) of 13 hummingbird species. We excluded Ruby-throated hummingbirds and broad-tailed hummingbirds from Perc ranking.

A high degree of specialization in resource use produces modules (*Dormann & Strauss, 2014*). We found a low degree of modularity, which denotes low resource specialization and random interactions (*Dormann & Strauss, 2014*), due to low resource specialization (plant-hummingbird) in this community (*Sonne et al., 2020*). Meanwhile, our results indicate that hummingbird species' dominance status and plants' nectar volume were the main traits that explain the separation of species into modules. The top species in the dominance hierarchy fed mainly on *Agave inaequidens* flowers, which are among the most abundant at our study site flowering plant species, and forming flower patches with higher energy availability (*López-Segoviano, Bribiesca & Arizmendi, 2018*). Thus, the modules are aggregated in sets of interacting species, reflecting that these species are linked more tightly together than species in other modules (*Olesen et al., 2007*; *Dormann & Strauss, 2014*). In Mexico, biogeographic origin is one of the main factors that separates hummingbird modules at the national scale (*Martín González et al., 2018*), and along an altitudinal

gradient at regional scale the distribution of plant and hummingbird species separate modules (*López-Segoviano et al., 2021*).

Some studies propose that dominant behavior can influence the preferences of hummingbird species for floral resources (*Stiles & Wolf, 1970*; *Sandlin, 2000*; *Maglianesi et al., 2014*; *López-Segoviano, Bribiesca & Arizmendi, 2018*), which can affect the rates of flower visitation that may be reflected in the hummingbird-plant network. Our study revealed that hummingbird species' specialization was related to their rank in the dominance hierarchy of the hummingbird community, in contrast to the findings of *Ramírez-Burbano et al. (2022)*, where the dominance hierarchy had a negligible effect on determining hummingbird specialization in the visitation network of feeders and plants in gardens. The authors attribute the lack of influence of the dominance hierarchy on specialization in that even non-dominant species can use the best food resources as territory intruders (*Justino, Maruyama & Oliveira, 2012*; *Ramírez-Burbano et al., 2022*). In addition, the hummingbird community of this study in Colombia is morphologically complex, with some non-territorial and specialized hermit species (*Stiles, 1975*; *Feinsinger & Colwell, 1978*). Consequently, morphological matching was the most important determinant of pairwise interactions (*Ramírez-Burbano et al., 2022*). It should be noted that in studies in the tropics, the non-territorial hermits were the most specialized species (*Maglianesi et al., 2014*, *2015*). Hermits' complementary bill morphology allows them access to almost exclusive resources, at the same time as it provides plants the best vectors for their pollen (*Stiles, 1975*; *Wolf & Stiles, 1989*; *Maruyama et al., 2014*; *Maglianesi et al., 2014*). In the temperate forest in Mexico, hummingbird species do not have exclusive resources; rather, they defend the floral resources they depend on for energy (*Sandlin, 2000*; *Rodríguez-Flores & Arizmendi, 2016*; *López-Segoviano et al., 2018*). Aggressive dominance allows the dominant species to access the best food resources (*Stiles & Wolf, 1970*; *Wolf, Stiles & Hainsworth, 1976*; *Sandlin, 2000*; *Rodríguez-Flores & Arizmendi, 2016*; *López-Segoviano, Bribiesca & Arizmendi, 2018*). Hence, the most dominant hummingbird species have the strongest interaction with the plant species whose flowers provided the highest energy floral reward as *A. inaequidens* (*López-Segoviano et al., 2018*).

It has been reported that hummingbirds' bill traits influence their degree of specialization in the hummingbird-plant interaction networks (*Dalsgaard et al., 2009*; *Maglianesi et al., 2014*, *2015*; *Tinoco et al., 2017*; *Sonne et al., 2019*; *López-Segoviano et al., 2021*). Our results showed that hummingbirds' $d'$ was related to bill length but not bill curvature. *Maglianesi et al. (2014)* in Costa Rica found that bill curvature has an important influence on hummingbirds' degree of specialization as well as resource use and niche partitioning in hummingbird assemblages (*Maglianesi et al., 2015*). However, in Northwestern Mexico, the bill curvature of the hummingbird community is less than in tropical regions. For example, in this study, Mexican Violetear had the greatest curvature, which can be very low compared to some species in tropical regions (*i.e.*, Hermits). Meanwhile, in our study the longer-billed and more specialized hummingbird species (Rivoli's Hummingbird and Blue-throated Mountain-gem) preferred to feed on

*A. inaequidens* flowers. These flowers have generalized floral traits with a small and straight corolla that is morphologically accessible to all hummingbird species at the study site. Morphologically specialized hummingbirds should avoid visiting plants with generalized floral morphologies (*i.e.*, short, straight floral corollas) to prevent competition with hummingbirds with generalized morphologies (*Maglianesi et al., 2015*; *Sonne et al., 2019*). In our study, the dominant species took the best resources by aggressively competing for them, regardless of floral traits. It seems that in this case, it is more important for hummingbirds to forage on flowers with high nectar volume than flowers with well-matched morphology (*Stiles, 1976*).

Our results showed that the core generalist species (White-eared and Rufous hummingbirds) were the less dominant hummingbird species. Contrary to what was described in an ant-plant network (*Dáttilo, Díaz-Castelazo & Rico-Gray, 2014*), in which the central core ants are generalists and are competitively superior to peripheral and submissive ant species. White-eared and Rufous hummingbirds had a medium bills and were the most abundant hummingbird in the region (*López-Segoviano et al., 2018*). In other hummingbird-plant networks, the hummingbird species' generalization was described as related to their abundance (*Simmons et al., 2019*; *del Arizmendi et al., 2021*; *López-Segoviano et al., 2021*). Likewise, the core generalist species are mainly associated with the abundance of the species (*Miele, Ramos-Jiliberto & Vázquez, 2020*; *Vitorino et al., 2022*).

Although dominance hierarchy shows some associations with assemblages between plants and hummingbirds (*López-Segoviano, Bribiesca & Arizmendi, 2018*), this pattern does not reflect in the interaction frequencies. The plant-hummingbird interaction frequencies were driven by abundance-based processes (neutral hypothesis; *Vázquez et al., 2009*), and dominance hierarchy may act as forbidden links. In this sense, *Sazatornil et al. (2016)* propose that the role of niche-based processes that structure interactions between mutualistic partners can be much more complex than previously established. In the study region, the three plant species in which the interactions are concentrated were also the more abundant. Therefore, many of the interactions were focused on these plant species, and even the two more abundant hummingbird species (with lower dominance) visit the more abundant plant species with low caloric content. Future studies will help to clarify the contributions of the dominance hierarchy to the structure and assembly of hummingbird-plant communities by incorporating data from other sites and communities.

It is essential to highlight that this study includes a temporal window of hummingbirds and plants' possible interactions. The ecological conditions, such as forage availability and predation risk, can be variable in time and across space (*Mayor et al., 2009*). Although we were able to record all of the hummingbird species that have been reported in the region (*López Segoviano, 2018*; *López-Segoviano et al., 2021*), the feeding behavior of hummingbirds and the structure of the network may change throughout the year (*Márquez-Luna et al., 2018*; *Bustamante-Castillo, Hernández-Baños & del Arizmendi, 2020*), and the aggressive organization and specialization level of hummingbirds can vary

depending on resource availability (*Justino, Maruyama & Oliveira, 2012*; *Rodríguez-Flores & Arizmendi, 2016*; *Tinoco et al., 2017*).

## CONCLUSIONS

Species abundance was the main driver of interactions in the plant-hummingbird assemblage (in accordance with the neutrality hypothesis). This may be explained by the sampling period restricted the appearance of forbidden links and did not reflect the importance of phenological overlap. Within this network, aggressive dominance of hummingbird species determines the separation into three different interaction modules. Thus, the relationship between hummingbird species, and their level of aggressive dominance determine niche partitioning among species (*López-Segoviano, Bribiesca & Arizmendi, 2018*) and was reflected in the hummingbird-plant assemblages. Our results suggested that feeding behavior may play an important role in North America's hummingbird-plant networks. Aggressive dominance and bill length of hummingbird species seems to be determinants of the hummingbird specialization of the hummingbird-plant network. Further studies are needed in more diverse communities and throughout the year, including the role of feeding behavior as a variable in the hummingbird-plant network (*i.e.*, *Ramírez-Burbano et al., 2022*). This study helps us understanding the importance of aggressive hummingbird behavior and the abundance of floral resources in the network of hummingbird-plant interactions.

## ACKNOWLEDGEMENTS

We are grateful for the comments and suggestions provided by Edith Villa-Galaviz. We acknowledge Ana Melisa Fernandes, Ann Hedrick, and the anonymous Reviewer for helpful comments and suggestions on an earlier manuscript. We are grateful for access to facilities at the study site provided by Ejido Forestal El Palmito. The authors thank Rafael Bribiesca, Oscar Félix, Alejandra Miranda, Ana Rubio, Pedro López and José Carlos López for field assistance.

### Funding

This work was supported by a grant from the UNAM-DGAPA-PAPIIT program (Project key: IN221920 and IN213223) to MCA. Gabriel López-Segoviano received the postdoctoral grant awarded by the Dirección General de Asuntos del Personal Académico (DGAPA) of the Universidad Nacional Autónoma de Mexico for the research carried out at the Escuela Nacional de Estudios Superiores Campus Morelia. The funders had no role in study design, data collection and analysis, decision to publish, or preparation of the manuscript.

### Grant Disclosures

The following grant information was disclosed by the authors:
UNAM-DGAPA-PAPIIT program: IN221920 and IN213223.

Dirección General de Asuntos del Personal Académico (DGAPA) of the Universidad Nacional Autónoma de Mexico for the research carried out at the Escuela Nacional de Estudios Superiores Campus Morelia.

## Competing Interests

The authors declare that they have no competing interests.

## Author Contributions

- Gabriel López-Segoviano conceived and designed the experiments, performed the experiments, analyzed the data, prepared figures and/or tables, authored or reviewed drafts of the article, and approved the final draft.
- Maribel Arenas-Navarro performed the experiments, analyzed the data, prepared figures and/or tables, authored or reviewed drafts of the article, and approved the final draft.
- Laura E. Nuñez-Rosas performed the experiments, analyzed the data, prepared figures and/or tables, authored or reviewed drafts of the article, and approved the final draft.
- María del Coro Arizmendi conceived and designed the experiments, analyzed the data, authored or reviewed drafts of the article, funding acquisition, and approved the final draft.

## Data Availability

The raw data are available in the Supplemental File.

## Supplemental Information

Supplemental information for this article can be found online at http://dx.doi.org/10.7717/peerj.16245#supplemental-information.

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
