# Peer review of "Implications of dominance hierarchy on hummingbird-plant interactions in a temperate forest in Northwestern Mexico"

_PeerJ, doi:10.7717/peerj.16245_

## Round 0.1 · original submission · Major Revisions

Please revise your manuscript, paying particular attention to the criticisms of the reviewers.

·

Basic reporting

In general, everything is clear and professional. However, I would like to see the interaction network, if possible, and maybe next to this, the species organized by ranking according to the Bayesian analysis (Adams, 2005).

Experimental design

For the aggressive interaction part, I recommend adding what you consider as a win or a loss and what type of aggressive interactions you consider (vocalizations, fights, chasses).

Also, for the hummingbird abundance estimations will be important to have further explanations about how you did this and avoided to count the same individual several times (how do you recognize the individual that did vocalizations is not the same that passed by flying).

Validity of the findings

I strongly recommend using the Bayesian analysis by Adams (2005) to assess dominance rankings. The DS index has some troubles with community data and does not give any certainty values on those rankings. Authors may be analyzing hierarchy with no sufficient data. Please use the Bayesian analysis to be more sure about these hierarchies and only use species with sufficient data.

Reviewer 2 ·

Basic reporting

The study investigates ecological factors associated with interaction frequencies in a Mexican hummingbird-plant network. The authors use the information on species abundances, morphological traits, feeding behaviour, plant energy content and phenologies to predict the pairwise interaction frequencies between each species of hummingbird and plant. Following the neutrality hypothesis, the result highlights species abundance as an important explanation for species interactions. Moreover, the individual bird species' specialization levels reflected partly long bill lengths and aggressive feeding behaviour.
Overall, I find the study interesting, relevant and well-written. Generally, we have poor information on hummingbird-plant interactions outside the mesic tropical regions, which has caused problems for previous studies of large-scale gradients in network structure and the ecological mechanisms influencing interaction frequencies. Hence, I find the study well-placed and contributing to close a long-existing knowledge gap. That said, I have some issues with aspects of the experimental design, analyses and presentation of the results that I believe should be addressed more thoroughly before publication.
The study differs from many published networks by sampling a smaller fraction of the year. The experimental design ensures that missing interactions in the network are not caused by species having non-overlapping phenologies. However, sampling a fraction of the year affects the relative importance of the ecological factors for the species' interaction frequencies. As the authors mention in the discussion section, the short sampling season explains why phenological overlap appears to be unimportant. For this reason, I cannot see how you can include phenology as a candidate factor in the first place. I would expect the sampling scheme to cover an entire production-active season to evaluate the implications of phenological mismatch. Even though the authors address this caveat in the discussion section, I fear that people will read the manuscript differently, with a high risk of the results becoming cited incorrectly – i.e. abundances are more important than phenologies and morphological matching. For these reasons, I suggest the authors remove phenological overlap as a candidate factor tested against abundances and morphological matching.
Secondly, I have problems using a forbidden links model to describe morphological matching. Although this model is frequently used in the insect-plant network literature, it is less applicable when studying interactions between hummingbirds and plants. Firstly there are issues with the information on the birds' tongue lengths that is often missing but nonetheless crucial for the model and the results. Importantly, the hummingbird's tongue length is not inferable from the bill length alone. Consider reading the Rico-Guevara's doctoral dissertation (2014) entitled "Morphology and function of the drinking apparatus in hummingbirds", which discusses this subject in detail. The Authors mention that they add a tongue length to the bill length when constructing the morphological matching model, but they do not mention how this information was collected. Moreover, the forbidden links model makes some odd predictions about the hummingbird's resource use that often do not coincide with observations in the field. Notably, hummingbirds with the longest bills are predicted to be generalized and have the most mutualistic partners – as they have access to all flowers. Under this framework, Morphological matching contributes to network nestedness, not specialization (Vázquez et al. Annals of Botany, 2009, cited), which does not coincide with general results from the literature. Generally, we find that long-billed hummingbirds interact predominately with long-corolla plants. Vice versa, short-billed species interact most frequently with small flowers. For this reason, the hummingbird-plant network literature typically uses a morphological matching model following concepts from optimal foraging theory. Here, long-billed hummingbirds are expected to interact most with long-corolla plants, regardless of the variability in tongue length, due to foraging efficiency and flower handling time. Vice versa, short-billed species should have the most interactions with short-corolla plants. There are several ways to formulate such a model. Weinstein and Graham (2017) & Sonne et al. (2019) have two suggestions. I suggest the authors use the same model for morphological matching as most of the studies cited in this manuscript.

Line-specific comments:
LINE 26: Words appear to be missing in the sentence.
LINE 93-94: "The strength of species interactions plays an essential role in determining the specialization, modularity, and core-periphery status". In what way? Could you clarify the mechanism?
LINE 97:103: Some hummingbirds, notably the Bees are specialized in having very high interaction frequency per unit of time. The ability to quickly interact with large numbers of flowers could be an adaptation to exploit abundant but energy-poor resources. Let's say that calories per flower vary drastically between two hummingbirds' resource niches. The net energy gained per time unit might be the same among the two species if the hummingbird visiting the low-calorie flowers has a higher interaction frequency. Hence, could the results regarding nectar energy change depending on the species' ability to exploit specific flower types?
LINE 177-180: The sampling of species dominance hierarchy needs to be explained more thoroughly. How do you define a win and a loss in the data set
LINE 215-219: How did you compare the 100 stochastic networks with your empirical data? Delta, z-score or something else?
LINE 231: How did you obtain measures of the hummingbirds' tongue length? This variable cannot be infeered from the bill length alone, as documented in Rico-Guevara (2014)
LINE 368-374: I do not see how your results contradict Maglianesi regarding hummingbird bill curvature. Could the fact that you find no association between d’ and curvature be caused by curve-billed species simply being absent in the dataset? The Mexican violetear is the species in the dataset with the greatest curvature. Its bill curvature appears very low compared to the species occurring in the Costa Rican networks.
FIGURE 2: Some hummingbirds and plants seem to have zero or very few interactions. I suggest the authors use another colour palette to clearly present the species with low interaction frequencies. If some species have zero interactions, I would not expect them to appear in a modularity graph like this.

Experimental design

see above

Validity of the findings

see above

---

## Round 0.2 · Major Revisions

The reviewers indicate that their major concerns remain unsolved. Please address all concerns in your revision; otherwise, I will have to reject your manuscript.

·

Basic reporting

They put one interaction matrix but with no time data. For one of the new dominance methods they add (Elo-Rating), they should have this information.

Experimental design

I still have many concerns about the dominance estimation methods they use. I asked them to add a Bayesian method to estimate the dominance certainty of the species. However, they add Bayesian analysis to estimate steepness certainty.

I recommend two methods to have dominance certainty for each species; Adams 2005 and Cowan et al. 2022 (I put the links in the document attached). In this way, they can make assumptions and conclusions only with the species they have enough information (dominance interactions). If you check their interaction matrix, it has a lot of zeroes (missing information), making unknown many of the dominant relationships between species.

Validity of the findings

The conclusions are not well stated until the dominance relationship issues are not solved. They have many zeroes in that matrix to make assumptions about dominance relationships and the interaction with other factors.

I strongly advise adding these two methods of dominance certainty and working just with those species with high dominance certainty.

Reviewer 2 ·

Basic reporting

I thank the Authors for addressing my precious comments. I cannot comment on the added "Steepness estimation with Elo-rating" as I have no experience with these analyses. However, I have some remaining issues that should be addressed before publication, with some concerning the methodology.

• The descriptions of species dominance hierarchy are still not explained thoroughly. You included a description of a "winner" but not what a "loser" is. Moreover, I would expect to see these definitions in the section about dominance hierarchy, not in the "Hummingbird interactions" section. Please move the descriptions to the correct section

• Using the Euclidean distance used to determine hummingbird-plant morphological matching does not account for tongue length! Moreover, it is incorrect that Sonne et al. 2019 used this procedure. They used a standardized approach instead (the z score) to acknowledge that the bird's tongues affect their foraging preferences without having the data. In this model, hummingbirds with the longest bills have the highest probability of visiting plants with the longest flowers. Similarly, birds with the shortest bills would be expected most frequently visit plants with the shortest corollas. The obvious flaw with the Euclidean distance is that it completely ignores the role of tongue length. Consequently, it becomes problematic to state that morphological matching is unimportant.

• I sympathize with the author's approach of using Vázquez's analytical approach. However, it is not well suited for continuous explanatory factors. It is perfect when you have binomial data, for instance, morphologically matching pairs versus forbidden links. The problem with the analytical approach is that it converts continuous morphological matching variables (such as the Euclidean distance and the z-score) into probability data. This conversion is not straightforward as one clearly has to define what is a forbidden link where species have zero interaction probability (i.e. how different should the species morphologies be?). This definition is super difficult when lacking the tongue length information, and then we are back to the author's framework in the initial submission.

Consequently, researchers increasingly turn to Bayesian analyses to document morphological matching in plant-hummingbird networks (e.g. Sonne et al. 2019 and Weinstein & Graham 2017). The authors could choose to go down that road. Otherwise, I suggest the authors use the z-score from Sonne et al. 2019 or the Euclidean distance from Weinstein & Graham 2017 to calculate morphological matching. Then scale these values between zero and one. One corresponds to a perfect fit, and zero corresponds to a clearly defined forbidden link. That could be bill length < corolla length + X, where X is a constant. Vizentin-Bugoni et al. 2014 (cited) used a value of X = 0.33. This 0.33 value is arbitrary, which should be addressed as a caveat. I would probably try using several values to explore how the results change accordingly.

• I am surprised that the authors find such a low yet significant modularity value. Moreover, the proposed modularity structure depicted in Fig. 2 is not very convincing. Could you please provide the Z-score and p-value in the results section? The low modularity is not surprising and coincides very well with the literature (See Dalsgaard et al. 2011, along with the more recent papers from his group). That said, I suggest the authors tone down the interpretation of each module and remind the reader in the discussion section about the overall low degree of modularity.

Specific comments on the authors' response letter
"R: It refers to the strength of the relationship of one species with another; in this case, it would be the number of visits of a hummingbird to a plant. Thus, the plants that hummingbirds visited, and the intensity of their visits determine various metrics of the interaction networks. According to Dormann and Strauss (2014), the strength of links as quantitative information is more sensitive and specific to detect modules. Likewise, the strength of the interactions is an essential piece in the calculation of specialization (Blüthgen, Menzel & Blüthgen, 2006) and core-periphery in the networks (Martín González et al., 2020)."
Please help the reader and update the sentence with a clarification.

Experimental design

see above

Validity of the findings

see above

Additional comments

see above

---

## Round 0.3 · Minor Revisions

Please re-revise your manuscript to meet the concerns of Reviewer 2.

·

Basic reporting

no comments

Experimental design

no comments

Validity of the findings

no comments

Additional comments

I think that Adams' analysis gives another type of certainty (how much information we have -number of interactions- to assume a ranking order for that species) than the Perc certainty that is more a biological one (it does not matter how many interactions you recorded, it only matters how easily is to recognize the dominant from the subordinate). However, I feel more comfortable now that they are not considering species with a low number of interactions or assuming a dominance hierarchy with low data.

Reviewer 2 ·

Basic reporting

Thanks to the authors for attending my comments. I still have some concerns with aspects of the methodology. I anticipate these concerns could be solved by additional clarifications.

The method for predicting species interaction frequencies takes much work to dissect. Most importantly, it needs to be clarified what response variable is used in the Generalized additive model. Is it pairwise interaction frequencies or a cumulative number of interactions per hummingbird and plant species? The confusion increases when seeing the predictor variables that go into the same model. The mean of calories per flower species is a plant-related trait, hummingbird weight is a bird-related trait, and morphological matching is a trait related to pairs of birds and plants; ergo, three potential response variables. The authors use the term frequency of interaction but is that a species-level term, or does it refer to pairwise interaction frequencies? Such confusion makes it difficult to understand exactly what is happening in the modelling procedure and how to interpret the results.

Line-specific comments:

Line 65-68: ”In seasonal regions, for instance, phenological overlap may be particularly important because hummingbirds’ migratory behavior leads to high seasonal turnover in species composition”

Important in which respect? For network structure? Please explain the mechanism. Please avoid using “important” as a stand-alone word, as it does not provide with an explanation.

Line 69-71: “Trait matching between hummingbird's bill traits and flowers’ corolla traits are an important driver of hummingbird-plant assemblages (Wolf & Stiles 1989, Maglianesi et al. 2014a, Sonne et al. 2019).”

The cited studies address the interaction structure in hummingbird-plant communities. From such studies, we cannot deduct that trait matching has implications for assembly processes. Such link has not yet been documented for hummingbird-plant communities. Also, please replace the word important, with an explanation of what the mechanism is.

Line 258: add ”t” to the word “predic”

Line 262-265: “We fitted a negative-binomial GAM with gamma to 1.4 to prevent over-fitting, using 1D splines for the observed trait values per species of each group (hummingbirds and flowering plants) and their matching (Benadi et al., 2021)”

Please clarify what the gamma and D parameter refers to. Moreover, what is the justification for using a value of 1.4?

Line 265: “We used the proportion of abundances…” It is not clear what the proportion of abundances refer to. Number of individuals relative to what? Why not simply use the raw abundance values?

Line 268-273: “We built morphological match as a forbidden link model, based on hummingbird bill length (+tongue) and curvature and flower corolla length and curvature: first standardizing all trait variables to zero mean and unit variance, second, the morphological match was calculated as the Euclidean distance in traits between each hummingbird–plant pair (Weinstein & Catherine, 2017; Sonne et al., 2019)”
In fact, the model for morphological matching, as described here, is not based on forbidden links. As explained in my previous review, using Euclidean distances between traits implies that some interactions are more likely than others, but no interactions are impossible. Hence, consider revising the first part of the sentence.

Line 273: “We added 26 % to the bill's total length to account for the tongue.”
The authors use a standardized approach to calculate morphological matching, where all trait variables to zero mean and unit variance. Biologically that operation makes no explicit assumption about the bird’s tongue lengths and should give the same measure of morphological matching regardless of the tongue-length factor used (hence, the reason why the standardized approach was used originally in previous papers). Therefore, if using the standardized approach, I would avoid using the tongue length factor and explain why. Sonne et al. 2019 contains some discussion on the topic.

Experimental design

See above

Validity of the findings

see above

---

## Round 0.4 · accepted · Accept

Thank you for revising your manuscript to meet the concerns of all of the reviewers.